# PlasmidMaker is a versatile, automated, and high throughput end-to-end platform for plasmid construction

Behnam Enghiad[1,2,3], Pu Xue [1,2,3], Nilmani Singh [1], Aashutosh Girish Boob[1,2], Chengyou Shi [1,2], Vassily Andrew Petrov[1], Roy Liu[1], Siddhartha Suryanarayana Peri[1], Stephan Thomas Lane[1], Emily Danielle Gaither[1] & Huimin Zhao [1,2✉]

Plasmids are used extensively in basic and applied biology. However, design and construction of plasmids, specifically the ones carrying complex genetic information, remains one of the most time-consuming, labor-intensive, and rate-limiting steps in performing sophisticated biological experiments. Here, we report the development of a versatile, robust, automated end-to-end platform named PlasmidMaker that allows error-free construction of plasmids with virtually any sequences in a high throughput manner. This platform consists of a most versatile DNA assembly method using *Pyrococcus furiosus* Argonaute (*Pf*Ago)-based artificial restriction enzymes, a user-friendly frontend for plasmid design, and a backend that streamlines the workflow and integration with a robotic system. As a proof of concept, we used this platform to generate 101 plasmids from six different species ranging from 5 to 18 kb in size from up to 11 DNA fragments. PlasmidMaker should greatly expand the potential of synthetic biology.

[1] Carl R. Woese Institute for Genomic Biology, University of Illinois Urbana-Champaign, Urbana, IL, USA. [2] Department of Chemical and Biomolecular Engineering, University of Illinois Urbana-Champaign, Urbana, IL, USA. [3] These authors contributed equally: Behnam Enghiad, Pu Xue. ✉email: zhao5@illinois.edu

Plasmids are one of the most foundational tools for recombinant DNA technologies. Design and construction of plasmids from smaller DNA parts to form complex functional DNA molecules such as biochemical pathways and genetic circuits is essential in molecular biology and represents one of the key steps enabling the design, build, test, and learn (DBTL) cycle in synthetic biology[1,2]. During the past two decades, a variety of innovative methods for plasmid construction have been developed. Some of the most notable examples include sequence homology based methods such as Sequence and Ligation Independent Cloning (SLIC)[3], isothermal Gibson assembly[4], uracil-excision based cloning[5,6], and yeast homologous recombination[7,8], or restriction digestion based methods such as MASTER ligation[9] and Golden Gate assembly[10]. Despite the development of numerous DNA assembly techniques, due to different capabilities and limitations of each approach, assembly of complex plasmid DNA molecules often requires trial of multiple techniques to find the suitable assembly approach for the DNA of interest or utilization of a multi-step hierarchical assembly scheme. As a result, construction of plasmid DNA remains one of the most time-consuming, labor-intensive, and inflexible steps in the DBTL cycle of synthetic biology, hampering the speed and scale of performing complex biological experiments.

Several strategies have been developed to address this limitation by automating the construction of plasmid DNA[11–15]. For example, high throughput synthesis of transcription activator-like effector nucleases (TALENs)[13] was achieved using automated construction of plasmids via Golden Gate method. In addition, a web-based software tool combined with a DNA assembly protocol using the Type-IIS restriction endonuclease based Modular Cloning technique was automated for efficient production of DNA fragments[11]. Although these automated plasmid construction methods have shown high productivity with accuracy, due to utilization of restriction digestion based methods such as Golden Gate assembly or addition of computationally designed linkers to join fragments[15], these platforms still lack flexibility in construction of plasmids with virtually any DNA sequence. This lack of flexibility is mainly due to inherent limitations of restriction digestion based assembly methods such as presence of restriction enzymes' recognition sequences on the DNA of interest or addition of scar sequences to improve the efficiency of DNA assembly. Moreover, most of the progress on automated DNA assembly has focused on the "build" part, where the initial design and final confirmation of plasmids are discrete. Therefore, it is highly desirable to develop a DNA assembly strategy that can assemble any DNA sequence with high fidelity and robustness and create an end-to-end pipeline for automated plasmid construction.

In this work, we report a robust, versatile, and automated end-to-end platform for plasmid construction named PlasmidMaker that enables scarless construction of virtually any plasmids in a high throughput manner. To implement this platform, we first developed a versatile, scarless, parallel, robust, and accurate method for assembly of multiple DNA fragments using *Pyrococcus furiosus* Argonaute (*Pf*Ago) based artificial restriction enzymes (AREs)[16]. We then designed both frontend and backend software for customers to build their specific DNA fragments using a user-friendly web interface and for our technicians to collect essential information that is required for DNA assembly, respectively. Finally, we integrated the DNA assembly method and the software with a robotic system named Illinois Biological Foundry for Advanced Biomanufatcuring (iBioFAB) (Supplementary Fig. 1) to create a nearly fully automated workflow for PlasmidMaker. As a proof of concept, we constructed 101 plasmids across six different species (bacteria, yeast, plants, and

mammals) which involves $\sim 2 \times 10^4$ pipetting steps using this PlasmidMaker platform. Plasmids with sizes ranging from 5 to 18 kb were assembled from up to 11 DNA fragments with limited human intervention. Our method allows assembly of fragments with GC content as high as 77% and error-free assembly of plasmids as large as 27 kb including the ones containing multiple repeats from up to 10 DNA fragments.

## Results

**PlasmidMaker overview**. The overall end-to-end pipeline for PlasmidMaker is shown in Fig. 1. There are four major steps involved in the automated workflow for plasmid construction: (1) In the Design part, users can design plasmids by arranging DNA fragments in their preferred manner as well as search for common plasmids from a database to serve as their templates by using the frontend software. After an order is received, a technician can perform quality check by using the backend software. The sequences which pass the checking criteria can then be sorted into picklists and sent for construction. (2) In the Build part, according to the worklists generated, PCR plates for amplification of fragments are prepared. The Tecan FluentControl liquid handler is controlled by Momentum^TM Workflow Scheduling software and F5 robotic arm to dilute and mix primers, templates, guides, and master mix into every single well of the 96-well plate. (3) Also in the Build part, one-pot automated *Pf*Ago digestion, purification, ligation, and transformation of different plasmids are performed on the iBioFAB. (4) In the Test part, constructed plasmids purified by automated minipreps are verified through restriction digestion and gel electrophoresis. Correctly assembled plasmids are then re-cultured for making frozen stocks by the liquid handler. This high throughput pipeline allows completion of multiple plates of DNA assembly via standardized and integrated unit operations. Researchers only need to supervise the whole system while the programmed software and robotic arms implement the complicated experiments.

**PlasmidMaker development: I. DNA assembly method**. The DNA assembly method using *Pf*Ago-based AREs is shown in the bottom panel of Fig. 1. In the first step, linear DNA molecules including fragments created by polymerase chain reaction (PCR), linearized circular plasmids, or synthesized dsDNA are mixed in a one-pot reaction containing wild type (WT) *Pf*Ago enzyme, a mutant *Pf*Ago enzyme (*Pf*Ago H745D, a.k.a. *Pf*Ago*), and single-stranded DNA guides. The linear DNA fragments are designed such that they each share 24 bp sequence homology to their neighboring assembly fragments. The ssDNA guides are also designed such that after digestion, 5′ sticky ends of 12 nt length are created (the combination of *Pf*Ago enzyme and DNA guides will be referred to as *Pf*Ago/ARE hereafter). After digestion of all linear DNA molecules in a one-pot reaction, the digestion products are purified and assembled using a high-fidelity DNA ligase in the second and final step. The assembly products are then transformed into *E. coli* cells and the resulting colonies are checked for correct clones.

Previously we were able to demonstrate *Pf*Ago's capability to create small size user defined sticky ends (i.e., 2–5 nt) on dsDNA molecules[16]. Although small size sticky ends like the ends created by most type II restriction enzymes are routinely used in DNA ligation/assembly applications, due to their low levels of base pair hybridization as well as limitations in the number of orthogonal overhangs, small size sticky ends are not ideal for assembly of multiple large DNA molecules. As *Pf*Ago/AREs can theoretically create any size of sticky ends, we decided to test this capability by creating 5–12 nt sticky ends on two linear DNA molecules ends (Supplementary Fig. 2a). To determine the efficiency of *Pf*Ago

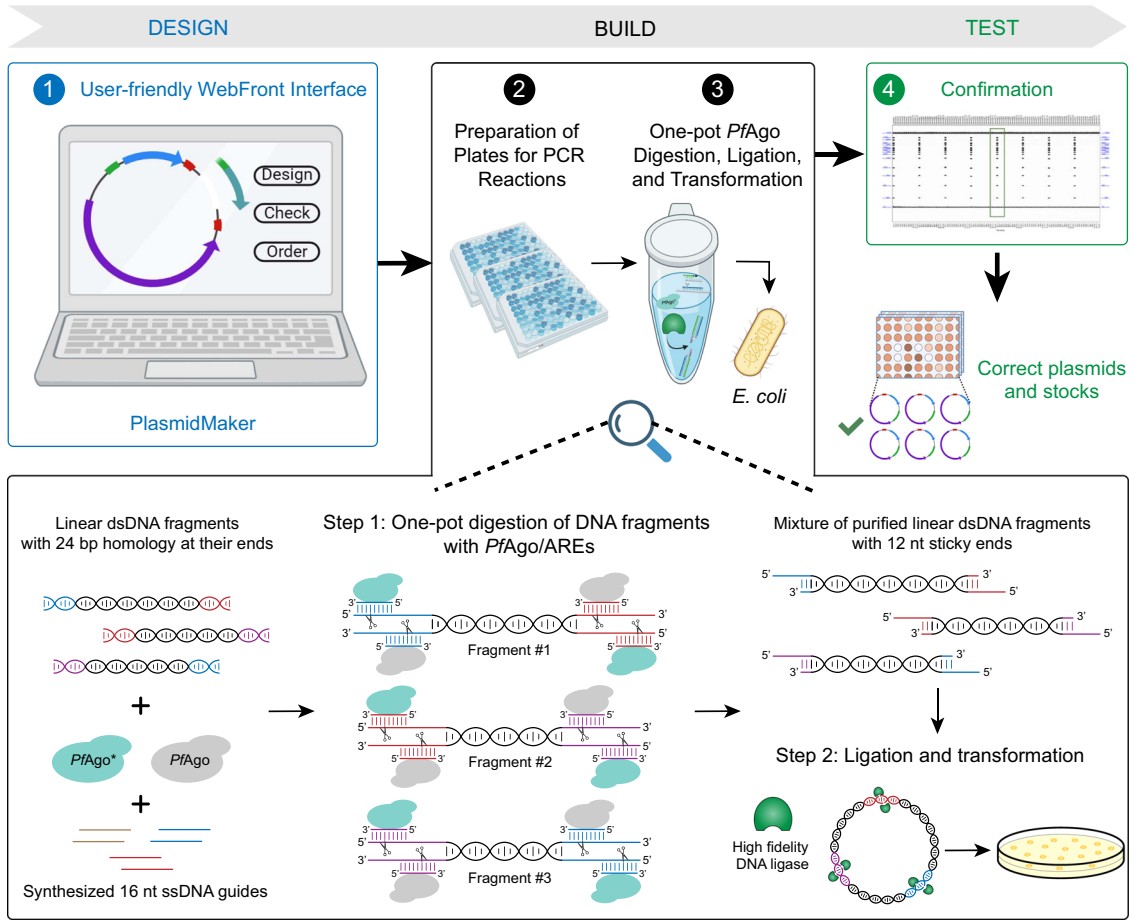

**Fig. 1 PlasmidMaker overview.** Design, Build, Test cycle for construction of plasmids from linear DNA parts. In the first step, a frontend is used to design and choose DNA fragments for assembly. After all the selected sequences passed the quality check, picklists (input for "Build" part) are generated for large-scale synthesis of primers and guides. The received oligos in either 96-well format or 384-well format are transferred into corresponding destination wells for PCR reactions to make specific DNA fragments via liquid handlers. Next, one-pot digestion, ligation, and transformation is performed inside iBioFAB to assemble DNA fragments. *Pf*Ago-based AREs are used for DNA assembly. In the first step, linear DNA molecules ends are digested with WT and engineered *Pf*Ago/AREs in a one-pot reaction. The AREs generate 5′ sticky ends of 12 nt length. After purification, the digested DNA molecules are assembled in vitro using a high-fidelity DNA ligase and assembly products are transformed into *E. coli* cells for screening. Finally, constructed plasmids (input for "Test" part) are checked and the correct plasmids are stocked using the robotic system.

cleavage on DNA ends for different sticky end sizes, we placed 24 bp of DNA homology at the ends of *amp* and *CrtI* genes by PCR amplification and performed *Pf*Ago cleavage using 5–12 nt guide DNA sets. Following digestion, purified digestion products were assembled using T4 DNA ligase. The efficiency of assembly product formation was used as a measure to determine *Pf*Ago's cleavage efficiency on DNA ends. As shown in Supplementary Fig. 2b, *Pf*Ago can effectively create 5–12 nt sticky ends on linear DNA molecules. Except for 10 nt sticky ends in which the first guide DNA can also ligate to the created sticky end after digestion (Supplementary Fig. 3), all the other sticky ends can be efficiently used for DNA assembly applications.

Other than advantages such as higher levels of base pair hybridization and increased probability for creation of orthogonal overhangs, when longer sticky ends are generated, *Pf*Ago/AREs demonstrate higher cleavage specificities. As shown in Supplementary Fig. 2a, when 5′ sticky ends of 5 nt are created, the two DNA guides jointly target a 17 bp of dsDNA sequence (a.k.a. *Pf*Ago/ARE recognition sequence). However, when the sticky end size is increased to 12 nt, the DNA guides jointly target a 24 bp of dsDNA sequence. As a result, *Pf*Ago/AREs creating longer sticky ends provide higher cleavage specificities. Since in the assembly of multiple DNA molecules, higher cleavage specificities lower the

probability of off-target cleavage and increase the success rate for acquiring the correct assembly products, we decided to choose *Pf*Ago/AREs generating 9 and 12 nt sticky ends for further characterization.

*Pf*Ago/AREs rely on dissociation of DNA strands at high temperatures for cleavage of dsDNA molecules[16]. One of the factors affecting strand dissociation is DNA GC content[17]. To test whether the GC content of *Pf*Ago/AREs recognition sequence can affect their cleavage efficiencies, we placed 10 randomly generated recognition sequences for both 9 and 12 nt AREs with different GC-contents and GC-distributions on the ends of two linear DNA fragments and analyzed DNA cleavage efficiencies by *Pf*Ago/AREs for each recognition sequence (Fig. 2a). We speculated that other than *Pf*Ago/AREs recognition sequence, the overall GC content of the whole DNA fragment might also play a role in *Pf*Ago/AREs cleavage efficiency. Therefore, we performed the same experiments on three sets of linear DNA molecules with three different GC content ranges. Previously, we created a mutant *Pf*Ago* which can utilize either $Mg^{2+}$ or $Mn^{2+}$ ions as a cofactor for efficient digestion of DNA molecules[18]. In our initial screenings, *Pf*Ago* demonstrated different sequence affinities compared to the WT *Pf*Ago enzyme. Therefore, we decided to also test the effects of GC content on *Pf*Ago*/AREs.

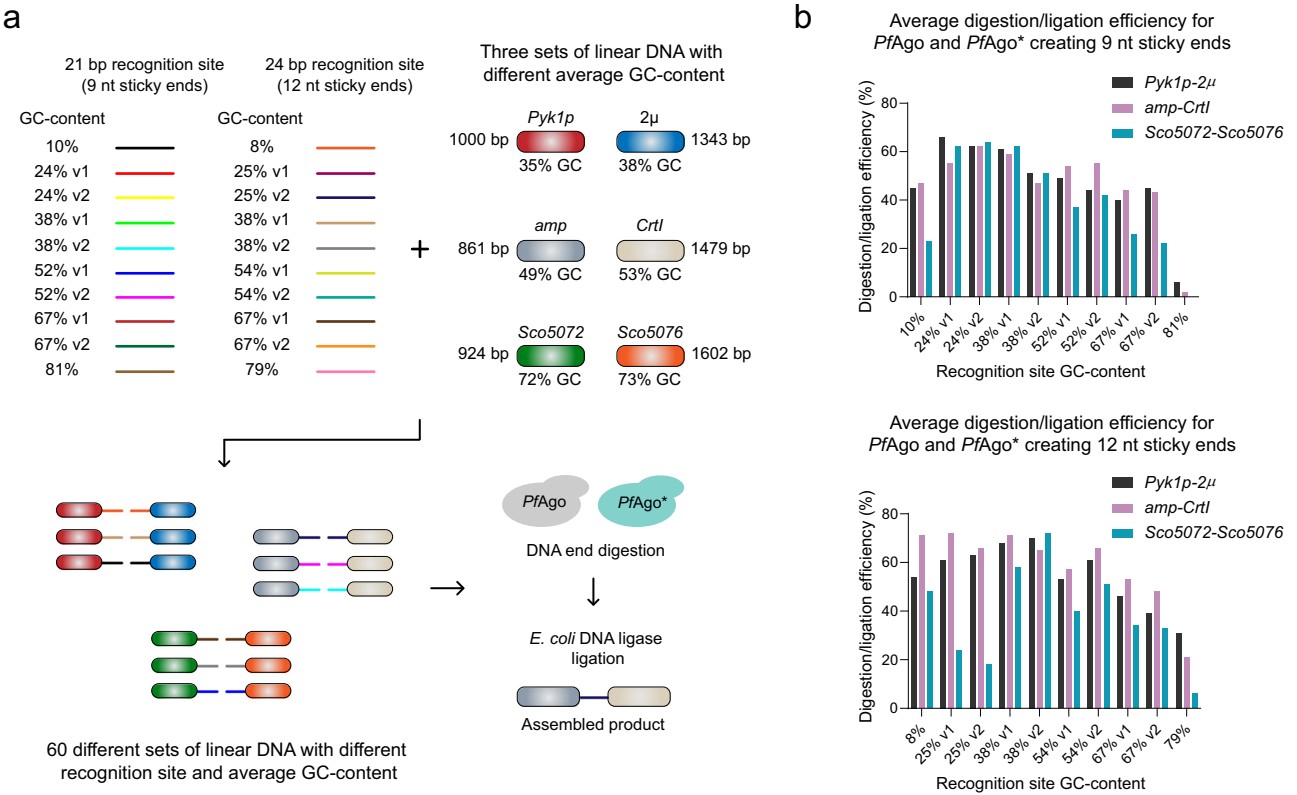

**Fig. 2 Analysis of the effects of *Pf*Ago/AREs recognition sequence GC content and GC-distribution as well as the fragments overall GC content on DNA assembly using *Pf*Ago/AREs. a** 10 randomly generated recognition sequences for both 9 and 12 nt AREs with different GC-contents and GC-distributions were placed on the ends of three sets of linear DNA with different overall GC content to create 60 sets of linear fragments. Each set was then digested by either WT *Pf*Ago or *Pf*Ago*/AREs creating 9 or 12 nt sticky ends and assembled by *E. coli* DNA ligase. v1 and v2 represent different GC-distributions. **b** Average cleavage/DNA assembly efficiency of both *Pf*Ago and *Pf*Ago*/AREs creating 9 or 12 nt sticky ends. Based on these results, *Pf*Ago/AREs can be programmed to cleave linear DNA ends with a wide range of GC-contents (0–75%). This data was used to help our guide design program to ensure efficient cleavage of DNA ends by *Pf*Ago/AREs (see Supplementary Text). Source data are provided as a Source Data file. The assembly efficiency analysis for each set was performed only once.

As shown in Fig. 2b and Supplementary Figs. 4a–c, except for cases where the AREs recognition sequence GC content was ~80%, both WT *Pf*Ago/AREs and *Pf*Ago*/AREs were able to cleave their target DNA molecules. As expected, WT *Pf*Ago and *Pf*Ago* demonstrate different sequence affinities. When the GC content of the recognition sequence is ~10%, WT *Pf*Ago demonstrates higher cleavage efficiency compared to *Pf*Ago*. On the other hand, if the GC content of the recognition sequence is higher than 50%, *Pf*Ago* cleaves its target at higher efficiency. While for DNA fragments with overall GC-contents of ~36% (i.e., *Pyk1p*-2μ) and ~51% (i.e., *amp*-*CrtI*), no significant difference in DNA cleavage/assembly efficiency was observed, both *Pf*Ago/AREs and *Pf*Ago*/AREs exhibited lower cleavage efficiencies when the overall DNA GC content was increased to ~73% (i.e., *Sco5072*-*5076*). In addition, creation of different length of sticky ends (i.e., 9 or 12 nt) and the recognition sequence GC-distribution did not have major effects on AREs' cleavage efficiencies. Taken together, these results indicate that *Pf*Ago/AREs can be programmed to cleave linear DNA ends with a wide range of GC-contents (0–75%) (Fig. 2b) and used in the assembly of multiple linear DNA fragments.

After testing the effects of both 9 and 12 nt sticky ends as well as a variety of DNA ligases on assembly of multiple DNA fragments (see Supplementary Text and Supplementary Fig. 5), we decided to use 12 nt sticky ends and HiFi *Taq* DNA ligase in the final version of our method and evaluated its performance.

To characterize the method's error rate, we designed a five-fragment assembly for a 7.2 kb plasmid (pAmp-ZeaX) harboring

a functional zeaxanthin pathway in *E. coli* (Supplementary Fig. 5a). As the resulting colonies harboring the full pathway produce yellow zeaxanthin, this plasmid allows quick assessment of DNA assembly fidelity based on colonies' colors. However, production of yellow zeaxanthin does not necessarily guarantee an entirely correct assembly without any sequence errors. To evaluate whether DNA assembly by *Pf*Ago/AREs would result in sequence errors between assembly junctions, we sequenced the five assembly junctions targeted by *Pf*Ago/AREs on pAmp-ZeaX plasmid for 14 randomly selected yellow colonies. Out of the 70 sequenced junctions, all junctions showed correct assembly without any sequence errors. To find out whether any of the white colonies harbor the full pathway but with sequence mutations, we analyzed purified plasmids from three white colonies by restriction digestion. Interestingly, one of the three plasmids showed correct digestion pattern for the complete 7.2 kb plasmid. After full plasmid sequencing, we found out that the lack of zeaxanthin production is due to four base pair deletions at the *CrtZ* gene stop codon. This section however is not targeted by any of the *Pf*Ago/AREs and all the five junctions targeted by *Pf*Ago/AREs showed correct assembly without any sequence errors. Since in plasmid pAmp-ZeaX, *CrtZ* and *CrtB* genes share 62 bp of sequence homology on their ends and they are amplified in one fragment, we speculate that this deletion occurs during the PCR reaction and due to the distance of mutation to the *Pf*Ago/AREs targets, it cannot be correlated to *Pf*Ago/AREs' cleavage. Consequently, these results indicate that

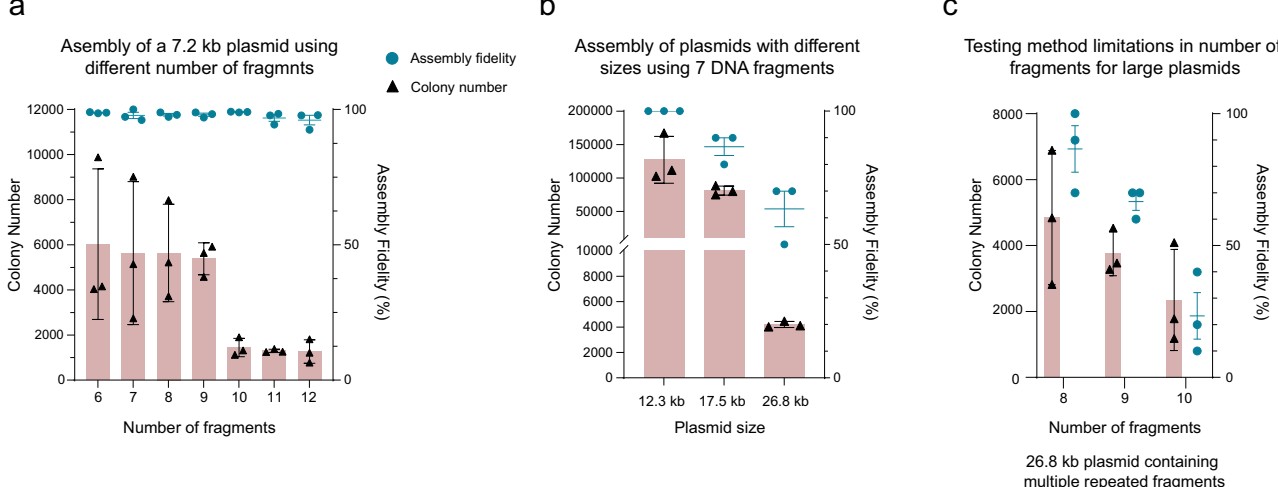

**Fig. 3 Characterization of *Pf*Ago/ARE based DNA assembly capabilities in terms of number of DNA fragments and overall plasmid size. a** Analysis of the effects of number of fragments on DNA assembly fidelity and total number of acquired colonies in assembly of the 7.2 kb pAmp-ZeaX plasmid. Assembly fidelity was calculated based on the ratio of yellow colonies to total acquired colonies. All experiments were performed in three biological replicates. **b** Effects of final product size on assembly fidelity and number of acquired colonies for assembly of plasmid DNA molecules with sizes ranging from 12.3 to 26.8 kb using seven DNA fragments. **c** Results for assembly of the 26.8 kb plasmid with 8, 9, or 10 DNA fragments. Based on these results, plasmid molecules with sizes up to 27 kb including the ones with sequence repeats can be efficiently assembled by *Pf*Ago/AREs generating 12 nt sticky ends from up to 10 DNA fragments. All experiments were performed in three biological replicates. For all graphs, error bars for colony number and assembly fidelity, show standard deviation (s.d.) and standard error (s.e.m) respectively. Source data are provided as a Source Data file.

*Pf*Ago/AREs offer exceptionally low error rate in assembly of multiple linear DNA fragments.

To evaluate the effect of total number of fragments on assembly fidelity and number of acquired colonies, we attempted to assemble the pAmp-ZeaX plasmid using 6 to 12 PCR-amplified linear DNA fragments (Supplementary Fig. 6a). By keeping the size of final assembly product constant, this strategy allows direct analysis of the effect of fragment numbers on both AREs cleavage and DNA ligation. As shown in Fig. 3a, we were able to successfully assemble the pAmp-ZeaX plasmid with up to 12 linear DNA fragments (a larger number of fragments was not tested). Strikingly, the DNA assembly fidelity was not significantly affected by the number of input fragments as all the assemblies exhibited higher than 95% fidelity. We speculate that this high assembly fidelity might be attributed to the high specificity of DNA ligation by HiFi *Taq* DNA ligase. As expected, increasing the number of fragments resulted in a lower number of acquired colonies. However, we were still able to observe a significant number of colonies (~1,000) for the 12-fragment assembly. Taken together, these results indicate that *Pf*Ago/AREs are capable of efficient and specific cleavage of multiple DNA fragments by using at least 24 different DNA guides in a one-pot reaction and the resulting cleavage products can be accurately assembled with a high-fidelity DNA ligase.

To analyze the effect of the final assembly product size on the assembly fidelity and the number of acquired colonies, we sought to assemble three different plasmids with varying sizes ranging from 12.3 to 26.8 kb using seven DNA fragments (Supplementary Figs. 6b–d). These plasmids included a 12.3 kb plasmid harboring an (*R,R*)-2,3-butanediol (BDO) pathway and a *gfp* gene[19], a 17.5 kb plasmid harboring the BDO and zeaxanthin pathways and a *gfp* gene, and a 26.8 kb plasmid harboring an n-butanol pathway as well as *gfp* and *mCherry* genes[20]. The 26.8 kb plasmid also included four 500 bp repeated *TEF1* promoter sequences. To minimize the effect of DNA size on transformation efficiencies and hence the number of acquired colonies[21], the assembly mixtures were introduced into *E. coli* cells by electroporation. For evaluation of the assembly fidelity, we picked 10 colonies from

each assembly and analyzed them by restriction digestion. As shown in Fig. 3b, both the assembly fidelity and the number of acquired colonies are inversely affected by the size of final product. However, despite the effects of size on DNA assembly, we were able to assemble the 26.8 kb plasmid using seven DNA fragments with ~63% fidelity and acquire ~4200 colonies.

To explore the possibility of increasing the number of fragments, we attempted to assemble the 26.8 kb plasmid using 8, 9, or 10 DNA fragments. As shown in Fig. 3c, we were able to assemble this plasmid with up to 10 DNA fragments. Taken together, these results suggest that plasmid molecules with sizes up to 27 kb including the ones with sequence repeats can be efficiently assembled by *Pf*Ago/AREs generating 12 nt sticky ends from up to 10 DNA fragments.

**PlasmidMaker development: II. Software**. To empower a broader set of users to use PlasmidMaker, we developed a web-based frontend software that is used to place orders to iBioFAB and a complete backend pipeline for automated assembly (Fig. 4a). We used industry-standard tools and practices for all phases of the software lifecycle. To facilitate rapid development, we selected the ReactJS framework for its widespread support for common open-source libraries and existing design templates. We selected the Django framework for backend processing for interoperability with existing laboratory software written in Python. The frontend is a fast and reactive website to design plasmids that can be sent through the automated workflow pipeline. Users must register to create an account that keeps track of their plasmids. In the main view, users can create new projects, which starts the process for plasmid construction. Once a project is selected, the project editor is opened, where the users either select from an existing library of plasmid fragments or upload custom fragments. We provide a text-based search and a sequence-based search to explore available fragments in the library. The main view includes a plasmid visualizer to display the final output, and a sequence visualizer to view which components can be placed in the project (Supplementary Fig. 7). Once the user has created an order of their liking, the fragments are annotated

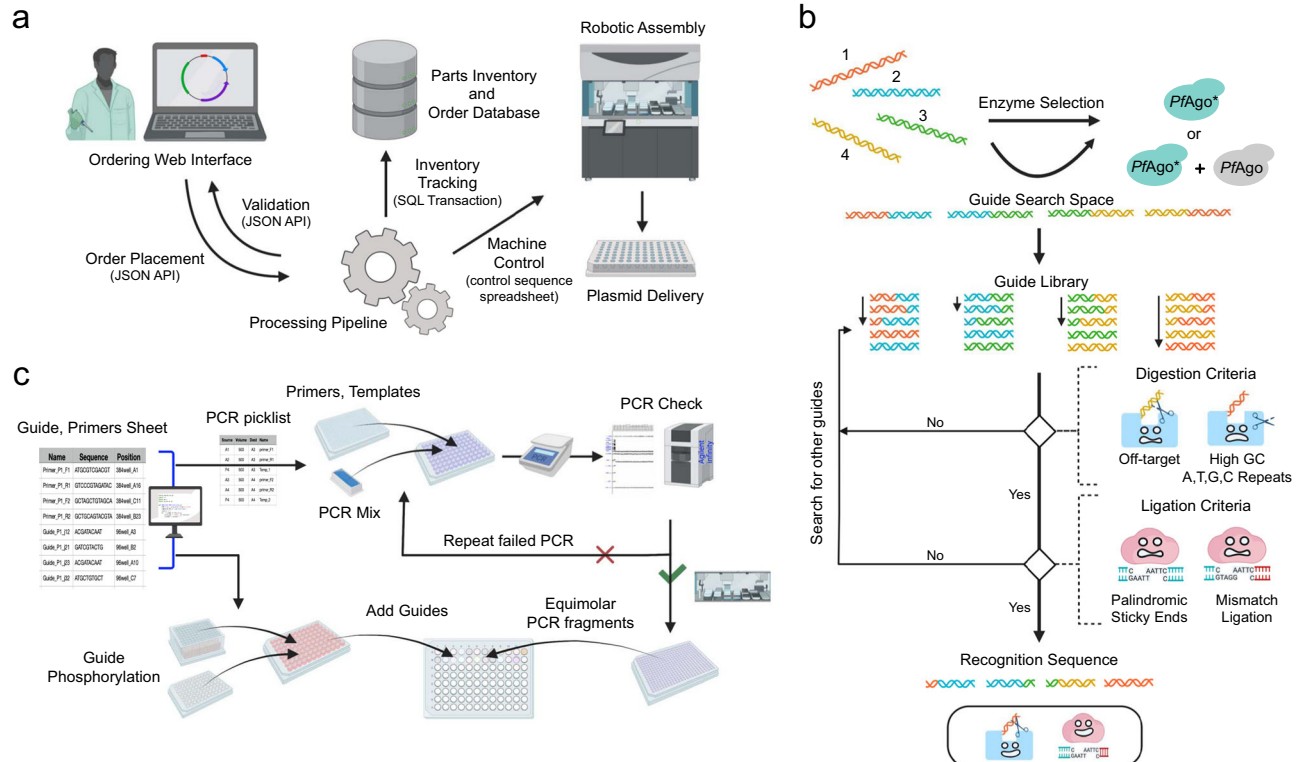

**Fig. 4 Overview of the workflow of the automated DNA assembly process. a** An online ordering system receives plasmid orders to be assembled. The received plasmid sequence is annotated and processed to generate primers, guides, and liquid handling worklists. **b** Algorithm for design of DNA guides and choice of enzyme for efficient *Pf*Ago-based assembly. Using annotated fragments of the plasmid as input, guide search space is created from the junction of the fragments. Based on the GC content of the fragments and guide search space, a suggestion is provided to use either WT *Pf*Ago and *Pf*Ago* or only *Pf*Ago*. The guide library created from the guide search space is scanned to identify 24 bp recognition sequences for high-fidelity plasmid assembly. The recognition sequence is finalized as soon as the guides satisfying the design rules for *Pf*Ago digestion and Hifi *Taq* ligation are obtained. **c** Workflow for generation of liquid handling worklists for *Pf*Ago-based plasmid assembly. Using reference csv file containing the location of primers/guides in 96-well/384-well plate for each plasmid, liquid handling steps are created for mixing primers and templates, mixing guides, followed by equimolar mixing of purified PCR fragments.

to obtain guides and primers. The order can also be submitted to iBioFAB for synthesis of the customized plasmid. Updates to the order after submission are tracked in real time through the projects tab. At each phase of an order, users receive updates as to the progress of their order. We use the React frontend library to serve performant webview renders to the client, and to ensure protection from cyber-attacks, an authentication system is implemented which obfuscates it from Cross-Site Scripting attacks through expirable cookies.

One of the challenges in designing a user interface that is intuitive to use is the layout of the controls and system state. To validate our design decisions, we incorporate a rolling-feedback method for collecting sentiment and experience data. After each iteration of the interface, we send requests to perform challenges to potential users of the frontend, and request that they assemble a specific plasmid in such a way that they will have to use all or most of the functionality that is available to them. After the challenge, we check whether they were able to successfully create the design, and we solicit feedback on their experience along with a video recording of their testing session. From the data collected, we can see what difficulties the user encountered or where they could have saved time if it was structured differently. This feedback is invaluable to the continuous improvement of the software. Common difficulties are resolved this way when the system state is not immediately obvious to the user. For example, a button click moves the window to a different tab and the user does not know how to return to the previous state.

To simplify and improve the fidelity of *Pf*Ago-based plasmid assembly, we developed a computational framework to generate design components given the annotated '.DNA' file (Supplementary Fig. 8). We first obtain the plasmid sequence, guide search space and the off-target library from the nucleotide sequence of the annotated fragments. The guide search space is defined as the region obtained by joining 40 bp from the end of the adjoining fragments for locating the guides. Based on the GC content of the fragments and the guide search space, the code provides suggestions on which *Pf*Ago enzymes to use for the assembly. The guide library is created using the guide search space of 80 bp and therefore, contains 57 recognition sequences of 24 bp to screen for. For each junction, a 24 bp recognition sequence (containing two guides of 16 bp each) is selected based on the digestion and ligation criteria (Fig. 4b). The digestion criteria include constraints based on the GC content, presence of G-quadruplex and more than 4 consecutive A's or T's and off-targeting to maximize the efficiency of creating desired sticky ends using *Pf*Ago and *Pf*Ago*. The ligation criteria are based on avoiding palindromic sticky ends and minimizing the mismatch ligation below a threshold score based on the mismatch fidelity profile of Hifi *Taq* ligase[22]. Once the viable guides are obtained, primers are then designed based on the position of the recognition sequence to include overhangs in the PCR amplicons for generating complementary 12 bp sticky ends (Supplementary Fig. 9). The length of the primer binding site is decided based on the GC content of the 18 bp binding region to ensure

amplification works across fragments of varying GC content. The primer binding site is further modified such that forward and reverse primers have $T_m$ values within 1 °C. The $T_m$ values of the primers are calculated using the 'calcTm' function of Primer3[23]. For assembling smaller fragments such as linkers and *E. coli* promoters (<80 bp), the workflow was modified to design primers for combining smaller fragments with the adjacent fragments via PCR overhangs and afterward, guides and primers are generated for assembly of the modified fragments.

To verify whether the plasmid assembly is correct, the script also generates the combination of restriction enzymes to use such that bands formed while performing gel electrophoresis are well separated and observable. The restriction enzymes can be selected from a list of commonly used restriction enzymes provided in Supplementary Table 1. We first check for the recognition sequence of the restriction enzymes and locate them in the plasmid sequence. Then, we look at the combinations of the restriction enzymes and discard the ones which do not satisfy the following criteria so as to observe well separated bands during gel electrophoresis: (1) Smallest band size ≥100 bp; (2) Largest band size ≤9500 bp; (3) If the difference between subsequent bands <A × band size, combination discarded where A is the accuracy of Fragment Analyzer and is set to 0.1 (Supplementary Fig. 10). If multiple '.DNA' files are provided in a folder, the script generates files for guides and primers for each plasmid assembly as well as a combined file for a list of restriction enzymes for correct assembly verification and repeated fragments across plasmids. We also address the problems in the assembly of the plasmid library through quality control scripts. All the details of the algorithm are explained in brief in the Supplementary Information.

We employed the Django Python framework to develop the backend server of the web application. Such a framework allows for seamless development of API endpoints, database management, and background tasks through a model-view-controller framework (Supplementary Fig. 11). The backend has a set of API calls that the frontend could call for necessary stored information from user data to custom plasmid information (Supplementary Table 2).

We used a PostgreSQL database to store experimental plasmids from a set of template plasmids, and important parts (promoters, genes, terminators) from the templates. With such storage of data, users can search genetic parts that they would like to add to their desired custom plasmids using the web interface. After an order is submitted, a background job managed by a Celery queue uses the plasmid validation and primer/guide design Python script to generate necessary primers and guides. Additionally, the Celery queue provides restriction enzyme combinations that are used to verify the assembled plasmid (Supplementary Fig. 11).

The automated DNA assembly requires picklist/worklist generation to smoothly integrate the liquid handling on different instruments on the iBioFAB (Fig. 4c). Using custom Python scripts and the output of the order queue, we generate picklists for various assembly steps. A custom Python script combines and removes duplicated DNA sequences from the primers/guides file generated from the PlasmidMaker order queue for one DNA assembly batch and formats it for ordering. Using the information sheet from the manufacturer, another Python script generates picklist to add water to the lyophilized DNA. A subsequent picklist dilutes and transfers primers to a separate 384-well plate, which will work as ECHO liquid handler source plate for PCR set up. The locations of primers/guides in 96-well/384-well plate for each plasmid is written in a csv file using a Python script and this file is used as reference file for creating liquid handling picklists during DNA assembly.

For setting up PCR, a Python script reads the reference csv files along with the location of PCR templates and creates picklist for mixing of primers and templates from 384-well plate to 96-well PCR plate in ECHO liquid handler. The Python script generates multiple PCR picklists for dividing the PCRs into multiple plates if the total number of PCRs exceeds 92. Each 96-well PCR plate has 92 PCRs and empty wells are used for DNA ladder for Fragment Analyzer run. Some DNA assemblies require insertions of <80 bp DNA fragments, which is achieved by designing extended PCR primers to add the inserts through PCR. A separate Python script creates the picklists for first round of PCR for adding <80 bp inserts. The PCR from first round is used as template for second PCR to generate the PCR fragments for assembly. The PCR products are analyzed using Fragment Analyzer and a Python script compares the expected and observed DNA sizes. When PCRs fail, a picklist is created combining all failed PCRs, followed by optimizing the PCRs. After all PCRs are successful, another Python script generates the picklist for guide mixing and phosphorylation. For digestion of PCR fragments with *Pf*Ago/AREs, a Python script generates picklist for equimolar mixing of purified PCR fragments and phosphorylated DNA guides. After the digestion, PCR fragments are purified, ligated, and transformed in *E. coli* cells for colony picking and verification of correct assembly.

**PlasmidMaker development: III. automation.** The whole automated workflow consists of three major modules: (1) Sample preparation through PCR and purification; (2) Automated DNA assembly and transformation; and (3) Plasmid purification, confirmation, and preparation of frozen stocks (Fig. 5a). For each building block in a module, a specific script was written in Momentum$^{TM}$ to execute that protocol. The detailed experimental procedures, inputs and outputs between each module, the involved equipment, and human interventions are summarized in Supplementary Figs. 12 and 13. In overall, this automated workflow can perform multiple parallel plasmid constructions within three days. Of note, the duration of 3-days includes the time-consuming steps for cell inoculation and overnight culture. If we only count the amount of time spent in each module every day, ~8 h is needed on average. Therefore, we are able to minimize the amount of labor involved in the automated workflow. Researchers only need to make decisions based on the experimental results and repeat modules if necessary.

The schematic overview of the three modules is shown in Fig. 5b. Firstly, we design and order the primers and guides for each plasmid using the backend software. Multiple liquid handling steps are then performed on Tecan FluentControl to hydrate the primers and guides into 100 µM by adding specific amount of water into each well according to our input worklists. The guides are saved for module 2 to set up the phosphorylation by Fluent. The primers are first distributed from 384-well source plate into 96-well destination plate by Echo liquid handler. The templates for DNA fragments amplification are also diluted into 1 ng per µL and 0.5 ng is added into corresponding wells in the destination plate by Echo. The plate with mixed primers and templates is then sent to Fluent again to make the master mix and combine all necessary parts with DNA polymerase for PCR reaction. In general, we use Echo to transfer liquid when the volume is less than 3 µL to ensure accuracy and precision. After we obtain all the PCR fragments and confirm their length, on deck one-pot *Pf*Ago digestion, ligation, and transformation are processed through the central F5 robotic arm controlled by Momentum$^{TM}$ software. Individual samples are cultured overnight in the Cytomat incubator for plasmid extraction after agar plating and colony picking. Finally, the purified plasmids are digested by specific restriction enzymes and checked by Fragment Analyzer (Fig. 5b). In addition, we developed two Python scripts

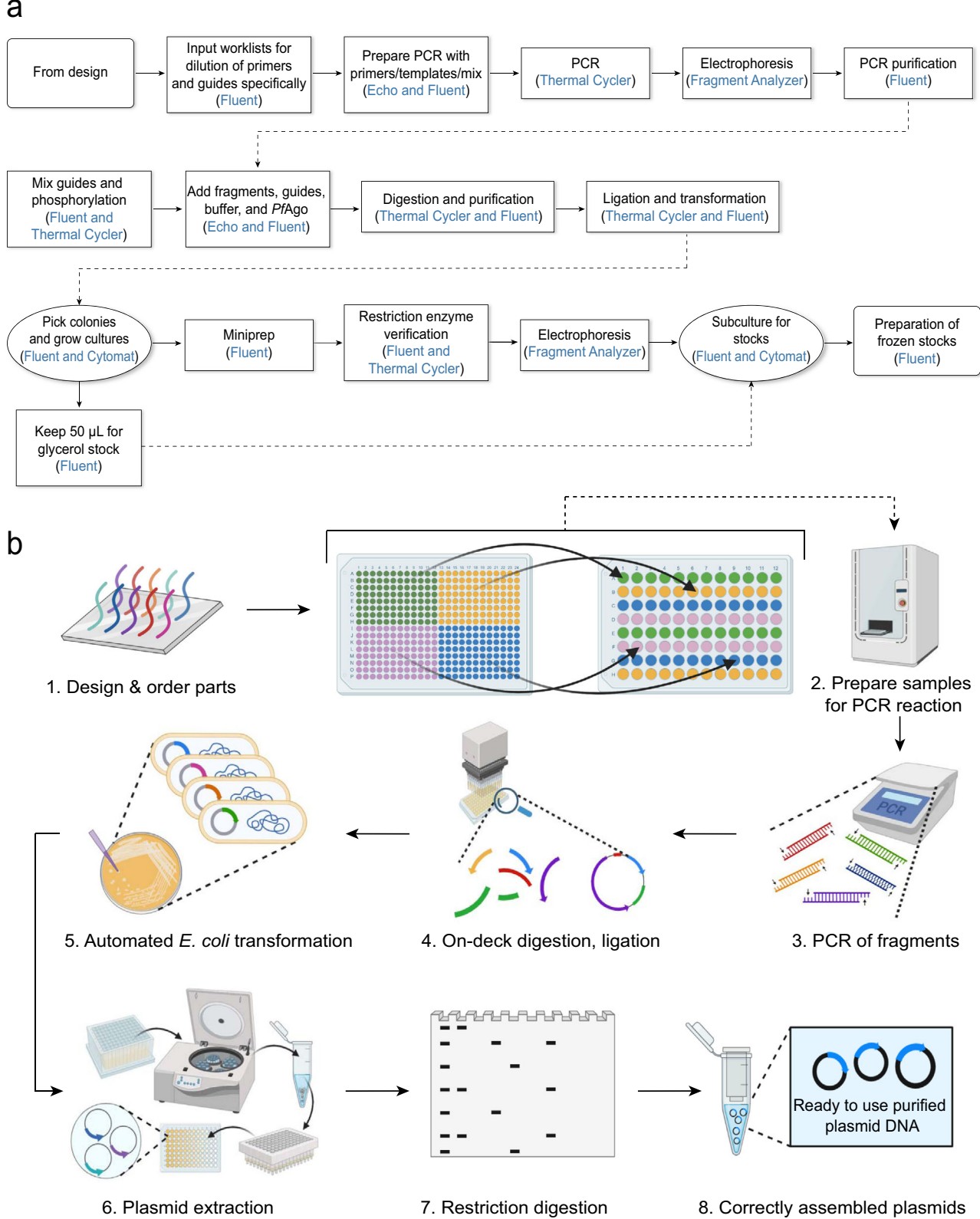

**Fig. 5 Overview of the automated plasmid construction workflow. a** A flowchart showing different instruments across iBioFAB connected through Momentum™ to orchestrate the plasmid construction process. The whole process starts from design of oligos and ends at preparation of frozen stocks of correctly assembled plasmids. The steps of experiments are listed in black character and the corresponding instrument are listed in blue. Rectangle shape: automated steps without incubation and shaking. Oval shape: automated steps with incubation and shaking. The detailed protocols are available in Supplementary Fig. 12. **b** Schematic overview for a closed automated plasmid construction workflow inside iBioFAB with all the detailed procedures and equipment listed.

| | E. coli | S. cerevisiae | I. orientalis | A. thaliana | HEK 293 T | S. griseoflavus |
|---|---|---|---|---|---|---|
| Plasmids constructed | 22 | 25 | 21 | 10 | 19 | 4 |
| Number of correct colonies/total colonies checked | 49/88 | 64/100 | 45/144 | 27/150 | 63/76 | 19/32 |
| Plasmids size range | 6–10 kb | 9–10 kb | 11–12 kb | 12–18 kb | 5–9 kb | ~13 kb |
| Number of fragments in total | 111 | 200 | 131 | 81 | 103 | 27 |
| Fragment numbers range | 3–6 | 8 | 5–10 | 5–11 | 4–8 | 5–8 |
| Fragments size range | 876–2715 bp | 225–2297 bp | 389–2767 bp | 293–3109 bp | 399–2370 bp | 920–3053 bp |
| Total primers/guides | 496 | 801 | 546 | 408 | 432 | 123 |
| Unique primers/guides ordered | 238 | 235 | 321 | 218 | 301 | 108 |

**Table 1 Summary of results for construction of 101 plasmids across bacteria, yeast, plant, and mammalian cells. Source data are provided as a Source Data file.**

in the confirmation module: one for quick identification of common restriction enzymes among different constructed plasmids, and the other for matching the predicted and actual band sizes efficiently.

**PlasmidMaker validation: automated construction of 101 plasmids.** To validate the PlasmidMaker platform, we sought to construct 101 plasmids for a wide variety of commonly used model organisms including *Escherichia coli*, *Saccharomyces cerevisiae*, *Issatchenkia orientalis*, *Arabidopsis thaliana*, *Human embryonic kidney (HEK) 293 T cells*, and *Streptomyces griseoflavus*. The summary of all the constructed plasmids including their assembly fidelity, plasmids size, number of fragments in total, fragments size range, and primers/guides involved is listed in Table 1. This table demonstrates the robustness and capability of PlasmidMaker in construction of a variety of plasmid with different sizes and number of fragments. Restriction digestion checks by Fragment Analyzer for the 101 constructed plasmids is available in Supplementary Fig. 14. In overall, we were able to construct all the designed plasmids with fidelities ranging from 18 to 90% with the plasmid with the largest size (18 kb) and number of fragments (11 fragments) showing the lowest fidelity. We were also able to construct plasmids containing multiple fragments with GC content as high as 77%. Because of the uniqueness of *Pf*Ago/AREs, our DNA assembly method provides the most versatile approach for construction of plasmids containing multiple repeats and secondary structures without any hinderance from "forbidden" DNA sequences. The detailed information and features of each individual plasmid are shown in Supplementary Table 3.

**Discussion**

In this study, we developed a versatile and automated end-to-end platform for design and construction of virtually any plasmids in a high throughput manner. This platform is built upon an innovative *Pf*Ago/AREs-based DNA assembly strategy and seamless integration of a computational infrastructure (i.e., frontend and backend) with a fully-integrated robotic system (i.e., iBioFAB).

An ideal method for assembly of DNA molecules should allow quick, scarless, error-free, sequence independent, and parallel assembly of multiple DNA molecules in a predetermined order and with high efficiency[2]. While DNA assembly strategies based on native type IIs restriction enzymes such as Golden Gate assembly allow parallel assembly of multiple DNA molecules in a predetermined order with high efficiency and low error rate, they generally suffer from two major limitations: (1) All DNA fragments used in assembly are required to be free of the restriction enzyme recognition sequences. As restriction enzyme recognition sequences are typically very short (~6 bp), this requirement limits their application in assembly of large DNA molecules due to low

probabilities of finding restriction enzymes which do no not cleave the DNA molecules in the middle; (2) restriction enzymes only generate short (~4 nt) sticky ends. Due to the small space of possible orthogonal 4 nt sticky ends that do not result in mismatch ligation and at the same time demonstrate high ligation efficiencies, assembly of multiple DNA fragments is generally performed using sets of standardized, predefined sticky ends[24,25] which can lead to assemblies with scar sequences. Unlike restriction enzyme based DNA assembly methods, long overlap homology based methods such as SLIC, SLICE[26], or isothermal Gibson assembly are not limited by the presence of "forbidden" DNA sequences in the middle of DNA fragments. However, homology-based DNA assembly methods are not truly sequence independent. As most homology-based methods rely on creation of long ssDNA ends through exonuclease treatment, the success rate and efficiency of these methods drops significantly if the generated long ssDNA ends can form stable secondary structures or contain sequence repeats[27–29]. In addition to exonuclease treatment, some of the homology-based methods such as iso-thermal Gibson assembly also rely on end-filling using DNA polymerase enzymes which can result in creation of junctions with sequence errors. As described elsewhere[30], in some cases, the error rate for junctions assembled by Gibson assembly can be as high as 100%.

Compared to native restriction enzymes, *Pf*Ago/AREs offer programmability as well as much longer recognition sequences. As a result, assembly of DNA molecules using *Pf*Ago/AREs is not limited by the presence of "forbidden" sequences on the DNA of interest. Additionally, *Pf*Ago/AREs can also be programmed to create varying sticky end sizes. Generation of longer sticky ends compared to native restriction enzymes (i.e., 12 nt in this study) results in substantially higher probabilities for acquisition of orthogonal sticky ends as well as possibility of utilization of thermophilic DNA ligases which offer remarkable ligation specificities. Unlike exonuclease enzymes, *Pf*Ago/AREs can create defined sticky end sizes. Compared to long ssDNA ends generated by exonuclease treatment (>20–40 nt), creation of defined sticky ends with moderate size (i.e., 12 nt) significantly reduces the probability of DNA secondary structure formation. *Pf*Ago/AREs also offer capabilities in assembly of fragments containing sequence repeats at their ends. As the location for cleavage of DNA molecules by *Pf*Ago/AREs is predefined, if this location (i.e., 24 bp) does not lie within the repeated sequence, presence of the repeated sequence would not interfere with assembly by *Pf*Ago/AREs. Lastly, as presented in this study, due to use of defined sticky ends and DNA ligation, DNA assembly with *Pf*Ago/AREs can be essentially performed with 0% error rates.

Traditionally, multiple cloning procedures are performed manually, which is labor-intensive and error-prone. Specifically, the possibility of human induced errors becomes very high when one trying to clone a large batch of plasmids, where numerous pipetting steps with tiny volumes of materials are required. With

automation of the cloning process, the assembly time is decreased tremendously as well as errors introduced by researchers can be avoided. In this work, $\sim 2 \times 10^4$ pipetting steps and transfer volume down to 2.5 nL are included to complete the automated plasmid construction. It will be very challenging to perform all these repetitive procedures manually with both precision and accuracy.

To establish an automated end-to-end platform for plasmid construction using *Pf*Ago/AREs-based DNA assembly, we developed a computational infrastructure consisting of a user-friendly frontend for plasmid design and a backend that streamlines the workflow and integration with our iBioFAB. The computational framework enables robust and high throughput construction of plasmids with high fidelity by incorporating complex criteria such as minimizing off-target sites and reducing the chances of mismatch ligation between the fragments which are strenuous to optimize manually. It also features the first bioinformatics algorithm for guide DNA selection of Argonaute proteins. The digestion efficiency of *Pf*Ago/AREs can be further increased by incorporating design rules for position-dependent base preference of *Pf*Ago enzyme or its variants[31]. Insights can be drawn from the computational model for predicting the sequence-specific activity of Cas9 variant[32,33] and applied to *Pf*Ago/AREs. Moreover, the ligation criteria can be improved by profiling mismatch ligation specifically for 12 nt sticky ends for HiFi *Taq* ligase.

One key challenge of building a fully automated DBTL cycle is smooth integration of the user frontend, the technician frontend, the software backend, and the robotic hardware-end. Users of the web interface are likely to be familiar with plasmid design and assembly technique in general but need to be able to design plasmids without knowledge of innovative *Pf*Ago/AREs. To facilitate a simple design process, the frontend uses validation of all user input to ensure compatibility with the assembly process. After we receive the requests for plasmid assembly from researchers on the PlasmidMaker user frontend, the backend software will perform BLAST[34] to search for sequences and annotate them. Meanwhile, the technician will generate the picklist for ordering according to the quality check results from software-end. If no primers and guides are found, a redesign of the assembly and/or an iterative check will be required. After we get the parts for assembly, the robotic hardware-end will perform the experiments following the inputs from the Design part. Ideally, no human intervention is needed in these steps, but if anything goes wrong in the middle of this cycle, the error will be recorded in our database and the continued run must be stopped. The role of the researcher is to monitor the whole workflow and respond quickly once an error occurs. Therefore, rational decision makings and expertise in automation are preferred for technicians before running the whole system.

One of the major challenges for high throughput assembly is robust integration of liquid handling steps over different instruments for different molecular biology assays. While worklists can be manually generated for each instrument, it is highly error-prone and labor-intensive. Here, we have developed an end-to-end Python module to generate all required worklists for DNA assembly on iBioFAB. The Python module takes the primer/guide information sheet as input and generates liquid handling worklists for various assays on iBioFAB such as PCR, guide mixing, phosphorylation, equimolar mixing of PCR fragments and *Pf*Ago/AREs digestion. The versatility of the worklist generation lies in the fact that it creates all necessary worklists for different reactions before we begin the assembly and works under many different assembly conditions.

Finally, to limit the human intervention in the automated workflow for plasmid construction, it is important to confirm the correct PCR amplification, the number of colonies being generated, and the digestion pattern at the end of each module, respectively (Supplementary Fig. 14 and Supplementary Table 3). Based on the results from the construction of 101 plasmids, the rate-limiting step is the PCR amplification of the fragments to be assembled. The success rate of PCR depends on many factors such as GC content, primer dimerization, primer multiple binding sites, secondary structures, length of fragment, annealing temperature, and DNA polymerase. In general, we could obtain more than 90% correctly amplified fragments after the PCR reaction in module 1. For the incorrect ones, we had to set up different conditions for troubleshooting, such as trying different DNA polymerases, using gradient annealing temperatures, adding GC enhancers, and adding dimethyl sulfoxide (DMSO) to the PCR master mix. If the PCR still fails, new primers need to be designed and new quality check procedures need to be conducted. Therefore, it is useful to develop a standardized, repeatable protocol for troubleshooting so as to improve the overall assembly efficiency. The other problem we faced during this workflow is the generation of false positive colonies with only circular backbone template plasmids. We solved this by adding *Dpn*I and incubating for 2 h after the PCR reaction and before purification in module 1 to get rid of the template structures. We also performed the Plasmid-Safe protocol after the ligation step in module 2 to improve the chance of getting true positive colonies.

In conclusion, PlasmidMaker is a powerful platform consisting of advanced software and hardware for rapid construction of virtually any plasmids in a high throughput manner. It addresses a critical bottleneck in basic and applied biology and should greatly accelerate the development of synthetic biology for biotechnological and biomedical applications.

## Methods

**Bacterial strains and reagents.** *E. coli* strain NEB10β (New England Biolabs, MA) was used for all cloning experiments. Restriction enzymes, T4 polynucleotide kinase, T4 DNA ligase, T7 DNA ligase, *E. coli* DNA ligase, *Taq* DNA ligase, HiFi *Taq* DNA ligase, ATP, and Q5 DNA polymerase were purchased from New England Biolabs, MA. Plasmid-safe ATP-dependent DNase was obtained from Lucigen, WI. DNA purification buffers PB and PE were purchased from Qiagen, Germany. All DNA oligonucleotides were ordered from Integrated DNA Technologies (Coralville, IA). All DNA guides were ordered unphosphorylated and the phosphorylation reaction was performed by T4 PNK enzyme prior to DNA cleavage.

**Expression and purification of WT *Pf*Ago and *Pf*Ago\* proteins.** The expression plasmids for WT *Pf*Ago (pET28a-*Pf*Ago) and *Pf*Ago\* (pET28a-*Pf*AgoH745D) were transformed into *E. coli* KRX (Promega, WI) following manufacturer's protocol. The strains were cultivated overnight at 37 °C in 5 mL of LB medium supplemented with 50 μg/mL kanamycin. Following overnight incubation, 2 mL of culture was transferred into 400 mL of Terrific Broth containing 50 μg/mL kanamycin in a 2 L baffled flask and incubated at 37 °C until the $OD_{600}$ of 1.2–1.5 was reached. The cultures were then cold shocked by incubation in an ice bath for 15 min and protein expression was induced by addition of isopropyl β-D-1-thiogalactopyranoside (IPTG) and L-rhamnose to final concentrations of 1 mM and 0.1% (w/v), respectively. Expression was continued by incubation at 30 °C for 20 h. Cells were harvested by centrifugation at $5,000 \times g$ for 10 min and after removing the supernatant, the cell paste was stored at −80 °C until purification. For the purification step, frozen cells were thawed at room temperature and then fully resuspended in 30 mL of buffer I (20 mM Tris-HCl pH 8.0, 1 M NaCl). The resuspended cells were lysed by sonication for 10 min pulse time (30% amplitude, 5 s on, 5 s off). The solution was then centrifuged 3 times at $20,000 \times g$ at 4 °C for 10 min periods and the supernatant was used for purification using fast protein liquid chromatography (FPLC) and 1 mL Strep-Tactin Superflow high capacity column (IBA Lifesciences, Germany). After equilibration with five column volume (CV) of buffer I, the column was loaded with the supernatant and washed with 10 CV of buffer I. The proteins were eluted with 10 CV of buffer II (20 mM Tris-HCl pH 8.0, 1 M NaCl, 2.5 mM *d*-desthiobiotin) and concentrated with Amicon Ultra-15 50 kDA filtration units (MilliporeSigma, MA). The purified proteins were diluted to ~1 mg/mL concentration using storage buffer (20 mM HEPES pH 7.0, 300 mM NaCl, 15% (v/v) glycerol) and the aliquots were stored at −80 °C.

**Characterization of WT *Pf*Ago and *Pf*Ago\* sequence affinities**. The *Pyk1p*, 2 μ, *amp*, *CrtI*, *Sco5072*, and *Sco5076* fragments were amplified by designed primer sets in 50 μl reactions using Q5 DNA polymerase following manufacturer's protocol. The PCR products were run on agarose gels and purified by Zymoclean Gel DNA Recovery Kit (Zymo Research, CA) following manufacturer's protocol. Before digestion of fragments by *Pf*Ago enzymes, the two DNA guides used for each set were each diluted to 100 μM concentration. Next, 5 μL of guide #1 and 5 μl of guide #2 were mixed in a 20 μL phosphorylation reaction containing 2 μL of 10x T4 DNA ligase buffer, and 2.5 μL of T4 PNK. The mixture was incubated at 37 °C for 1 h followed by 65 °C for 30 min and cooled to 4 °C.

*Pf*Ago digestion of linear DNA sets was carried out in a 25 μL reaction containing 500 ng of total target DNA (fragment #1 and fragment #2 with equimolar ratio), 2.5 μL of phosphorylated guides mixture, 12.5 μL of 2x *Pf*Ago reaction buffer (40 mM HEPES pH 7.0, 300 mM NaCl, 2 mM MnCl₂), and 0.5 μL of *Pf*Ago or *Pf*Ago\* enzymes. After complete mixing of the reaction components by pipetting, the mixture was incubated at 70 °C for 15 min, 85 °C for 10 min, followed by slow cooling (0.1 °C/s) to 10 °C. Following digestion, 125 μL of buffer PB (Qiagen) was added to the mixture and after complete mixing, the solution was transferred into a Zymo-Spin I column (Zymo Research, CA) and centrifuged at 21,000 × g for 30 s. After discarding the flow-through, the column was washed by addition of 400 μL buffer PE (Qiagen) followed by centrifugation and discarding the flow-through. This wash step was repeated one more time. Finally, the column was placed into a clean 1.7 mL centrifuge tube and 10 μL of ddH₂O was directly added to the column matrix. After sitting at room temperature for a minimum of 5 min, the column was centrifuged to acquire the purified digestion products.

Assembly of the digestion products was performed in a 10 μL reaction containing 5 μL of the purified digestion products, 1 μL of 10x *E. coli* DNA ligase buffer, and 0.5 μL of *E. coli* DNA ligase. The mixture was incubated at 25 °C for 2 h, 65 °C for 30 min, followed by cooling to 10 °C. The assembly products were then ran on agarose gels and visualized by Gel Doc XR + system (BioRad, CA). The band intensities for the assembly product and each DNA fragment were measured by the ImageJ software and the DNA cleavage/ligation efficiency for each set of fragments was calculated based on the following formula:

$$\frac{band\ intensity\ for\ the\ assembly\ product}{Combined\ band\ intensities\ for\ fragment\ \#1,\ fragment\ \#2,\ and\ the\ assembly\ product}$$

**Assembly of multiple DNA fragments using *Pf*Ago/AREs**

*Guide DNA phosphorylation*. Prior to one-pot cleavage of DNA fragments with *Pf*Ago enzymes, the unphosphorylated DNA guides used for assembly were each diluted to 100 μM concentration and 4 μL of each guide was added to a PCR tube to create a mixture of DNA guides with equimolar ratios. After complete mixing of the guides, 10 μL of this mixture was used for guide DNA phosphorylation. Phosphorylation of DNA guides was carried out in a 20 μL reaction containing 10 μL of DNA guides mixture, 2 μL of 10x T4 DNA ligase buffer, and 2.5 μL of T4 PNK. The reaction mixture was incubated at 37 °C for 1 h, 65 °C for 30 min and cooled to 4 °C. This reaction mixture can be stored at −20 °C if the guides are not going to be used immediately.

*Preparation of DNA fragments amplified by PCR*. The linear DNA fragments used in all performed assemblies were amplified by the designed primer sets in 50 μl reactions using Q5 DNA polymerase following manufacturer's protocol. In cases where the template was a plasmid DNA with the same resistance marker as final construct, less than 2 ng of template DNA was used for PCR amplification to limit the number of false positive colonies. Following PCR, the products were run on agarose gels and purified by Zymoclean Gel DNA Recovery Kit (Zymo Research, CA) according to manufacturer's protocol.

*Pf*Ago/AREs cleavage of multiple DNA fragments*. One-pot cleavage of multiple DNA fragments was carried out in a 50 μL reaction containing 1 μg total target DNA (all fragments with equimolar ratio), 5 μL of the phosphorylated guide DNA mixture, 25 μL of 2x *Pf*Ago reaction buffer (40 mM HEPES pH 7.0, 300 mM NaCl, 2 mM MnCl₂), 0.5 μL of WT *Pf*Ago, and 0.5 μL of *Pf*Ago\*. The solution was fully mixed by pipetting and incubated at 70 °C for 15 min, 85 °C for 10 min followed by slow cooling at the rate of 1 °C/s until 10 °C was reached. For assembly of the high GC content plasmids using only *Pf*Ago\*, the cleavage reaction was performed using 1 μL of *Pf*Ago\*. The mixture was incubated at 70 °C for 15 min, 92 °C for 10 min followed by slow cooling until 10 °C was reached.

*Purification of Pf*Ago cleavage products*. Following one-pot DNA cleavage, the 50 μL reaction mixture was transferred into a 1.7 mL centrifuge tube. Next, 250 μL of buffer PB (Qiagen) was added to the mixture and after complete mixing, the solution was transferred into a Zymo-Spin I column (Zymo Research, CA) and centrifuged at 21,000 × g for 30 sec. The centrifuge flow-through was discarded and 400 μL of buffer PE (Qiagen) was added to the column. After centrifugation and discarding the flow-though, the PE buffer wash step was repeated one more time. The column was then transferred into a clean 1.7 mL centrifuge tube and 10 μL of ddH₂O was directly added to the column matrix. To ensure maximum elution of cleavage products, the column was placed at room temperature for at least 5 min. Next, the column was centrifuged to acquire the purified cleavage products.

*Ligation of cleavage products by HiFi Taq DNA ligase*. The DNA ligation was carried out in a 20 μL reaction containing 5 μL of purified cleavage products, 2 μL of 10x HiFi *Taq* DNA ligase buffer, and 1 μL of HiFi *Taq* DNA ligase. The mixture was incubated at 37 °C for 2 h and cooled to 10 °C until transformation. Based on our experiments, 0.5 μL of *Dpn*I restriction enzyme can also optionally be added to the ligation reaction to help with lowering the number of false positive colonies formed by carryover of PCR templates during DNA purification. However, we only used *Dpn*I in assembly of the 26.8 kb plasmid using 8, 9, or 10 DNA fragments.

*Plasmid-safe treatment of the assembly products (optional)*. Following ligation, 2.3 μL of 10 mM ATP and 1 μL of plasmid-safe DNase were added to the ligation mixture and the sample was incubated at 37 °C for 90 min, 70 °C for 30 min followed by cooling to 10 °C until transformation. This plasmid-safe treatment step was only used in assembly of the 26.8 kb plasmid using 8, 9, or 10 DNA fragments.

***E. coli* transformation**. For chemical transformation, 5 μL of the ligation mixture was added to 50 μL of chemically competent NEB10β cells (New England Biolabs) and transformation was performed following the manufacturer's protocol. For electroporation, the ligation mixture was first dialyzed against ddH₂O using membrane filters (MilliporeSigma, MA, cat. no. VSWP02500) for 30 min at room temperature. 5 μL of the dialyzed mixture was then added to 25 μL of electrocompetent NEB10β *E. coli* cells (New England Biolabs), mixed gently and transferred to a 2 mm electroporation cuvette. Electroporation was performed using Gene Pulser XCell Electroporation system (Bio-Rad, CA) with 2500 V, 200 Ω, 25 μF condition.

**Analysis of plasmid assembly**. Following transformation, 1 mL of SOC medium was added to the cells and the suspended cells were transferred into 14 mL round-bottom Falcon tubes. The culture incubated at 37 °C for 60 min with shaking at 250 rpm. Different concentrations of cells were then plated on LB agar plates containing appropriate antibiotics (100 μg/mL carbenicillin) using cell spreaders and incubated at 37 °C until colonies appeared. To increase the intensity of the yellow color for colonies harboring the zeaxanthin pathway, after ~16 h of incubation at 37 °C, the plates were placed overnight at room temperature and the colonies numbers were calculated on the following day. For analysis of correct colonies, at least 10 single colonies were picked and grown at 37 °C in LB medium supplemented with appropriate antibiotics. The plasmid DNA was purified from the cultures using Qiaprep Spin Miniprep Kit (Qiagen, Germany) following the manufacturer's protocol. The purified plasmids were then digested by appropriate restriction enzymes and the samples were analyzed by agarose gel electrophoresis.

**Automation of plasmid construction**. iBioFAB was used to automate the assembly of DNA parts for the target 101 plasmids, including dilution of primers and guides, PCR reaction, gel electrophoresis, purification, guides phosphorylation, digestion, ligation, transformation, cell cultivation, plasmid extraction, and so on. SnapGene viewer software was used to analyze DNA files. SnapGene-reader was used to read all the 101 annotated plasmids files. The overall and detailed workflows of the experiments are shown in Fig. 5 and Supplementary Fig. 12, respectively. First, the primers and guides were diluted into 100 μM using pure water after the orders were received from IDT. The primers were transferred into 384-well plate from the 96-well IDT plate for module 1 sample preparation of PCR reaction. The guides were stored in 96-well plate for guide phosphorylation in module 2. The template plasmids were diluted to 500 pg-1 ng and added into 384-well plate as well. The primers and templates were then transferred from 384-well source plate into 96-well destination plate using Echo 550 liquid handler. In each well of the destination plate, it contained the specific primers and templates for one PCR reaction according to the input CherryPick csv. file. Meanwhile, the PCR reaction master mix was prepared by Tecan FluentControl liquid handler in a 96-well format. Since the volume required to dilute every sample into same final concentration was different, Flexible-Channel Arm (FCA) tips were used to perform the liquid transfer. This allowed efficient liquid handling with different volumes being transferred in each well. We also created a worklist in Fluent script for FCA transfer, which kept us on track in real-time. Next, the two 96-well plates were mixed according to our input worklist on Fluent to generate the final plate for PCR reaction (Fig. 4). The plate was then sent to sealer followed by thermal cycler using F5 robotic arm. After 35 cycles of PCR reaction, *Dpn*I was added to each well of the 96 samples simultaneously using Multi-Channel Arm (MCA) inside Fluent liquid handler. After another incubating in 37 °C for 2 hr, the plate was sent back to Fluent again to prepare for gel electrophoresis check of the fragment sizes. We used the Fragment Analyzer dsDNA 920 Reagent Kit (Agilent Technologies, Santa Clara, CA) to perform the automated gel electrophoresis analysis. Premade marker plate (30 μL/well + mineral oil overlay), inlet buffer plate (1000 μL/well), and 40 mL gel with 4 μL of intercalating dye were put into the corresponding trays in Fragment Analyzer. For preparation of the sample plate, 22 μL of 1x TE buffer was added to each well first by MCA, followed by 2 μL of the PCR product from the 96-well plate after *Dpn*I digestion. Ladders were added to wells A1 and H12. For these fragments, we chose the ladder with a calibrated range from 75 bp to 15,000 bp (part number: DNF-920-K0500). The total running time of automated gel electrophoresis took 1.5 h, then we could decide whether we obtained the correct band patterns or not using the ProSize data analysis software. After verification of the

sizes, we performed the PCR purification using the Zymo-96 DNA clean-up column-based Kit (D4018) on Fluent liquid handler. The DNA quantification step was done independently using the high throughput Lunatic Microfluidic system (Unchained Labs, Pleasanton, CA). Only 2 μL of eluted samples were needed to determine DNA concentrations.

The output csv. file of the DNA amounts was used to calculate the volume needed for each fragment in every assembly in order to obtain equimolar (1 μg of fragments in total). The fragments were mixed by Echo 550 and the guides were mixed and phosphorylated by Fluent and thermal cyclers, respectively (Fig. 4). The mixed PCR fragments and guides were then digested using *Pf*Ago and purified using the Zymo-96 clean-up Kit. Next, ligation of purified digested fragments was performed, followed by 1.5 hr incubation with plasmid-safe nuclease. The treated product was then transformed in NEB-10β *E. coli* competent cells and plated on LB agar plates and moved to Cytomat_2C2 by F5 robotic arm for overnight culture at 37 °C. On the second day, four colonies were picked from each of the plates using Pickolo colony-picker (SciRobotics, Israel) and inoculated in 96-deepwell plate with 1.4 mL of LB + antibiotic media in each well (more colonies were picked if we obtained false positive results after gel checking). The seed culture was grown overnight at 900 rpm and 37 °C. On the following day, 50 μL of culture was aspirated out first for preparation of frozen glycerol stocks. The rest of cultures were spun down at 15 min and 3,900 rpm using the Agilent centrifuge. Supernatants were removed and the cell pellets were kept for plasmid extraction. For miniprep, we first tried magnetic beads-based method using the MagJET plasmid DNA Kit (K2792) (ThermoFisher Scientific, Waltham, MA) on the Fluent. However, the concentration obtained was generally low with ~20 ng/μL in 30 μL elution volume. Then, we developed a vacuum-based plasmid extraction protocol named "TeVacS" on Fluent using the PureLink™ Pro Quick96 plasmid purification Kit (K211004A) (Invitrogen, Carlsbad, CA). Our customized protocol features the use of vacuum manifold for efficient removal of washing buffer and final elution of plasmid DNA into a 96-well plate placed on a 3D-printed holder. This alternative method took less than 90 min and yielded a DNA concentration of ~80 ng/μL in 50 μL elution volume. The purified plasmids were then digested using the selected common restriction enzymes listed in Supplementary Table 1 and analyzed by Fragment Analyzer. The correctly digested patterns of these plasmids were shown in Supplementary Fig. 14. Lastly, frozen stocks were prepared on Fluent using 40% glycerol and the overnight culture of correct clone in a 1:1 volume: volume condition. Either FCA tips or MCA tips were used depending on the number of plasmids created.

**Reporting summary**. Further information on research design is available in the Nature Research Reporting Summary linked to this article.

## Data availability
The data supporting the findings of this study are available within the article and its Supplementary Information files or uploaded through public repositories. If specific data is believed to be missing, that data is available from the corresponding author upon request. The plasmids for expression of *Pf*Ago enzyme and its mutant version are available through Addgene.

## Code availability
The custom codes for the frontend and backend software used in this study are available at Zenodo: https://zenodo.org/record/5812313 (https://doi.org/10.5281/zenodo.5812313). The source code to design guides, primers for *Pf*Ago-based plasmid assembly and select restriction enzymes for assembly verification on a local machine is available at Zenodo[35] and on Github at https://github.com/ibiofab/PlasmidMaker_GuideDNA. The web application is available on the following web address: https://biofoundry.web.illinois.edu/. The datasets used to populate the PostgreSQL database are available in the Supplementary Information files in a SnapGene format. The sequences can be converted into a Postgres format using the database scripts found in our code repository.

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

## Acknowledgements

This work was supported by U.S. Department of Energy award DE-SC0018420. Any opinions, findings, and conclusions or recommendations expressed in this publication are those of the author(s) and do not necessarily reflect the views of the U.S. Department of Energy. We thank Jonathan Chow Ning for providing the zeaxanthin plasmid template and helpful discussions about initial design of experiments. The online tool BioRender (biorender.com) was used to create Fig. 1's upper panel, Fig. 4a, b, and c, Fig. 5b, and Supplementary Fig. 8, 9.

## Author contributions

B.E. and H.Z. conceived and designed the study. B.E., P.X., N.S., A.B., V.A.P., and H.Z. wrote the manuscript. B.E. developed *Pf*Ago-based DNA assembly method as well as *Pf*Ago mutant, performed all the method development experiments, and devised assembly design rules. P.X., N.S., and C.S. designed and developed all the robotic automation workflow and performed the construction of all 101 plasmids using iBioFAB. E.D.G. and S.L. helped with initial robotic scripting on Tecan FluentControl liquid handler. N.S. created the Python modules for worklist generation for liquid handling on iBioFAB and helped develop quality control scripts. A.B. developed the computational model for guide DNA, primer design, and selection of restriction enzymes for plasmid assembly verification. V.A.P., R.L., S.P., and S.L. developed the web interface and software pipeline for enabling plasmid construction workflows.

## Competing interests

The authors declare no competing interests.
