## [Peer Review File · Nature Communications]

Reviewers' Comments:

Reviewer #1:

Remarks to the Author:

This paper describes an end-to-end workflow using both software and robotics to create plasmids. It includes computational and experimental components and places these in a clear design-build-test framework. The authors are aware of the challenges and needs in this space and have integrated this into their automated foundry, iBioFab.

This paper is admirable in that the authors attempt to cover the entire design flow. That comes at the expense of making the automated design and build algorithm descriptions vague and high level. It is hard to understand if there are novel data structures and algorithm advances in this work. I assume that the software is a lot of engineering effort with little computational advance (more like a commercial tool from someone like Teselagen). This is fine provided the software can be used easily by a large community. However, it is also hard to tell how the authors want the community to interact with the software. Should they be downloading it and getting it to work on in their labs? Is iBioFab going to take orders as a "cloud lab"? What is the proposed engagement model? Very few labs will have the iBioFab equipment. It would have been nice to see this positioned more as a frontend that will connect to several academic efforts with a procedure in place to get additional labs onboarded in the future. I would expect UIUC could have also put this in the context of deployment with Global Biofoundry Alliance Labs.

That being said the software workflow is very comprehensive and well-engineered (for an academic offering or otherwise) and should be promoted within the synthetic biology community. It would be nice to understand the software internal data structures and database schemas with those proposed by the larger community (e.g. SBOL).

This paper represents a real need but one that needs to be (1) distributed in such a way the software can be repurposed in other labs and (2) deployed in such a way that labs without this equipment can make use of these services.

Figure comments:

Figure 1 – Very nice figure overall. However, I would recommend the following:

1. Connect the outputs of each phase (design and build) with the inputs to the next (build and test). As it stands it is not clear how much is created automatically by the previous stage and how manual or automated the transfer of these inputs and outputs is.
2. You probably could integrate this all into one figure by putting part B into the Build section and just making this all one diagram of the DBT cycle.

Figure 2 – this figure heavily requires the caption. You could fix this with some simple high-level text of what each picture is showing. This figure begs the question about the issues with 10nt sticky ends which should at least be touched on.

Figure 3 – it is clear what analysis you did but what is the recommendation at the end of this? I would expect that you could highlight some important results and say how this information is used to inform the workflow going forward.

Figure 4 – again the results are there but the interpretation and its effect on the workflow are not. Is this information now in the tools and presented to the user during the design stage?

Figures 5 and 6 – these are nice figures that could benefit from:

1. Annotation on the arcs in the diagrams in Figure 5 with the file formats in part A that come out of the processing pipeline.
2. In Figure 5 part B it would be nice to understand visually when the loop between creating the guide library and going to the recognition sequence stops.
3. Figure 6 A – why are some steps rectangles and some ovals?
4. Figure 6 B is now another workflow that the reader has to reconcile with the other workflows shown. It would be nice to figure out how to relate it to the others, highlight the differences, and

show where the innovation is.

Table 1 – great to have these numbers. Is this good? Bad? What is the benchmark for comparison? I have to read the paper to find out more (this is expected but if that is the case, this might just be a good supplemental table).

Reviewer #2:

Remarks to the Author:

Authors report the development of a versatile, robust, automated end-to-end platform named PlasmidMaker that allows error-free construction of plasmids with virtually any sequences in a high-throughput manner. This platform consists of a most versatile DNA assembly method using *Pyrococcus furiosus* Argonate (PfAgo)-based artificial restriction enzymes, a user-friendly frontend for plasmid design, and a backend that streamlines the workflow and integration with a robotic system. They used this platform to generate 101 plasmids from six different species ranging from 5 to 18 kb in size from up to 11 DNA fragments within 3 days.

PlasmidMaker should expand the potential of synthetic biology.

The paper has provided a convenient method for plasmid construction. It can be accepted after following issues are addressed:

1. Automation is important for making plasmid library. Please list pro and con of the traditional and Plasmidmaker in the discussion;
2. Can the plasmidMaker assemble repetitive DNA sequences? This is generally very difficult for conventional plasmid library construction?
3. What are the largest and smallest sizes of plasmids that can be assembled by the PlasmidMaker?
4. Since soft- and hardwares are needed for the PlasmidMaker, I assume ordinary labs cannot use the plasmidmaker now, am I right?

We would like to thank the reviewers for their time and constructive comments. Below are our point-by-point responses to the reviewers' comments.

Responses to Reviewer #1:

This paper describes an end-to-end workflow using both software and robotics to create plasmids. It includes computational and experimental components and places these in a clear design-build-test framework. The authors are aware of the challenges and needs in this space and have integrated this into their automated foundry, iBioFab.

This paper is admirable in that the authors attempt to cover the entire design flow. That comes at the expense of making the automated design and build algorithm descriptions vague and high level. It is hard to understand if there are novel data structures and algorithm advances in this work. I assume that the software is a lot of engineering effort with little computational advance (more like a commercial tool from someone like Teselagen). This is fine provided the software can be used easily by a large community. However, it is also hard to tell how the authors want the community to interact with the software. Should they be downloading it and getting it to work on in their labs? Is iBioFab going to take orders as a "cloud lab"? What is the proposed engagement model? Very few labs will have the iBioFab equipment. It would have been nice to see this positioned more as a frontend that will connect to several academic efforts with a procedure in place to get additional labs onboarded in the future. I would expect UIUC could have also put this in the context of deployment with Global Biofoundry Alliance Labs.

That being said the software workflow is very comprehensive and well-engineered (for an academic offering or otherwise) and should be promoted within the synthetic biology community. It would be nice to understand the software internal data structures and database schemas with those proposed by the larger community (e.g. SBOL).

This paper represents a real need but one that needs to be (1) distributed in such a way the software can be repurposed in other labs and (2) deployed in such a way that labs without this equipment can make use of these services.

Responses:

We thank the reviewer for his/her many constructive comments. Below are our responses to each of his/her specific comments.

This paper is admirable in that the authors attempt to cover the entire design flow. That comes at the expense of making the automated design and build algorithm descriptions vague and high level. It is hard to understand if there are novel data structures and algorithm advances in this work. I assume that the software is a lot of engineering effort with little computational advance (more like a commercial tool from someone like Teselagen).

Response:

The algorithmic advance in this work includes the development of the first bioinformatics tool for guide DNA selection for efficient cleavage by Argonaute proteins. Another novelty of the work lies in the combined work which includes the development of the *PfAgo* based DNA assembly method, automation of DNA assembly and development of software to support the automation process. As the reviewer points out, combining the different algorithms using the

existing software engineering knowledge were indeed a lot of engineering effort. However, the in-house scripts were implemented with a focus on developing an alternative and easily accessible plasmid design and assembly tool.

This is fine provided the software can be used easily by a large community. However, it is also hard to tell how the authors want the community to interact with the software. Should they be downloading it and getting it to work on in their labs? Is iBioFab going to take orders as a “cloud lab”? What is the proposed engagement model? Very few labs will have the iBioFab equipment. It would have been nice to see this positioned more as a frontend that will connect to several academic efforts with a procedure in place to get additional labs onboarded in the future. I would expect UIUC could have also put this in the context of deployment with Global Biofoundry Alliance Labs.

Response:

We will make the primer/guide design tool for *PfAgo* based assembly as a public website and a free online tool. The research community will not need to download the software for their individual needs (although they can if they want). We will provide a detailed SOP as well as a video to help people understand how to use the online application. Hence, the research community will be able to design primers/guides for assembly and downloading all data for in-house use if they want. In addition, we will provide plasmid construction as a service, where the users can use the website to upload the plasmid sequences and request assembled plasmids from iBioFAB. Currently, we are rolling out the DNA assembly in different phases and we have started offering plasmid construction service to the researchers in the DOE Center for Advanced Bioenergy and Bioproducts Innovation (CABBI) since this February. We will further offer this service to all DOE bioenergy research centers at the end of this year, followed by releasing it for ordering from the broad research community. Finally, since we are the founding member of the Global Biofoundry Alliance, we will be more than happy to help other members to implement this PlasmidMaker pipeline in their institutions if they want.

That being said the software workflow is very comprehensive and well-engineered (for an academic offering or otherwise) and should be promoted within the synthetic biology community. It would be nice to understand the software internal data structures and database schemas with those proposed by the larger community (e.g. SBOL).

Response:

The software was written using commonly used programming languages that include Python, reactJS, Django Python, and PostgreSQL database. The overall design of the web frontend for *PfAgo* assembly and development of primer/guide design algorithm and liquid handling picklists did not use the SBOL schema.

Figure 1 – Very nice figure overall. However, I would recommend the following:

1. Connect the outputs of each phase (design and build) with the inputs to the next (build and test). As it stands it is not clear how much is created automatically by the previous stage and how manual or automated the transfer of these inputs and outputs is.
2. You probably could integrate this all into one figure by putting part B into the Build section and just making this all one diagram of the DBT cycle.

Response:

1. Since the outputs from the “Design” part are used as inputs for both “Build” and “Test” parts, it is hard to connect them directly in the figure due to limited space. To clarify the outputs and inputs, we revised the Figure 1 legend to include inputs for each specific section.
2. Figure 1b is the overall method for assembly of linear DNA fragments using *PfAgo*/AREs. We thought it is important to place this in the first figure so that the readers will be familiar with our core technology from the start. Since this Figure contains both design and build strategies, we would not be able to integrate it into the “Build” section only.

Figure 2 – this figure heavily requires the caption. You could fix this with some simple high-level text of what each picture is showing. This figure begs the question about the issues with 10nt sticky ends which should at least be touched on.

Response:

For each part, a text was added above to describe each part more clearly. For 10 nt sticky ends, a new figure was added to the Supplementary Information to provide detailed description since there is not enough space in the original figure.

Figure 3 – it is clear what analysis you did but what is the recommendation at the end of this? I would expect that you could highlight some important results and say how this information is used to inform the workflow going forward.

Response:

The purpose of this experiment was to analyze the ability of *PfAgo*/AREs to cleave DNA ends with any random sequence as well as their limitations. Based on these results, *PfAgo*/AREs can create sticky ends on recognition sites with a wide-range of GC-contents (0-75%) and higher than 75% GC-content seems to be their only limitation. We used these results for our guide design program which can be found in the Supplementary Information. However, because of word limitation, we did not describe it in detail in the main text. For more clarifications, we added new text to both the main text and the Figure 3 legend to specify the 0-75% GC-content as well as referring the readers to the Supplementary Information on how this data was used for guide design.

Figure 4 – again the results are there but the interpretation and its effect on the workflow are not. Is this information now in the tools and presented to the user during the design stage?

Response:

The interpretation of results was added to the Figure 4 legend. These experiments were designed to roughly characterize the limitations of our method in terms of plasmid size in combination with the number of fragments. Based on our current experience, we suggest the users to design plasmids with up to 15 DNA fragments. This number of fragments would work if the plasmid size is small (less than 10 kb). However, if the plasmid size is large and the number of fragments is also higher than 10, the plasmid construction might fail. Since accurate determination of the limitations in size and number of fragments requires many trials, at the moment we suggest the users to design plasmids with up to 15 DNA fragments regardless of plasmid size. Once more data is gathered, we can determine the limitations for the number of fragments in combination with the plasmid size.

Figures 5 and 6 – these are nice figures that could benefit from:

1. Annotation on the arcs in the diagrams in Figure 5 with the file formats in part A that come

out of the processing pipeline.

2. In Figure 5 part B it would be nice to understand visually when the loop between creating the guide library and going to the recognition sequence stops.

Response:

1. Figure 5a has been revised to include the annotations.
2. Figure 5b has been modified to include the flowchart symbols to help understand that the loop between creating the guide library and going to the recognition sequence stops as soon as the guides satisfying the digestion and ligation criteria are found. The stop is signified by the rectangle with rounded corners which in this case signifies if we satisfy both the ligation and digestion criteria, the search is terminated.

Figure 6 A – why are some steps rectangles and some ovals?

Response:

Oval means step includes Cytomat incubation and shaking, rectangle means normal automated steps without incubation and shaking. Changes have been updated in the Fig. 6a legend and Supplementary Fig. 8c legend.

Figure 6 B is now another workflow that the reader has to reconcile with the other workflows shown. It would be nice to figure out how to relate it to the others, highlight the differences, and show where the innovation is.

Response:

The purpose of this Fig. 6b is to present a closed automated system for plasmid construction with detailed procedures from the design part to the confirmation part and with all equipment being integrated to achieve the same goal. From this figure, readers would have a clearer idea of which equipment corresponds to which automated action and how different equipment are connected inside our biofoundry for plasmid assembly purpose. The difference between this figure with Fig. 1a is that Fig. 1a shows a broader overview of the entire DBT cycle without detailed equipment and procedures. The difference between this figure with Fig. 5a and 5c is that Fig. 5a and 5c focus more on the online ordering system connecting with the technician interface and on the steps for generation of worklists for downstream plasmid assembly, respectively.

Table 1 – great to have these numbers. Is this good? Bad? What is the benchmark for comparison? I have to read the paper to find out more (this is expected but if that is the case, this might just be a good supplemental table).

Response:

This data broadly demonstrates the capability of PlasmidMaker in constructing plasmids across many different species. We thought it is good to summarize the information of all the constructed 101 plasmids with species, fidelity, plasmid size, number of fragments, fragment size, and primers/guides being used. From this table, readers will have a clearer idea on whether PlasmidMaker can be used for their specific project. This table also demonstrates the robustness of PlasmidMaker in construction of a variety of different plasmid with different sizes and number of fragments.

Regarding a benchmark, there is not really a benchmark available for comparison. Mainly because our DNA assembly method is unique and provides the most versatile method for DNA

assembly which can construct plasmids containing multiple repeats and secondary structures without any hinderance from “forbidden” DNA sequences. Because of this uniqueness, we are not able to compare it directly to other strategies as they are unable to construct such plasmids with this level of versatility. Data with this scale is also not available for other methods. Because of the aforementioned reasons, we think this table is important to be placed in the main text because it summarizes the application part of our manuscript. However, if the reviewer still believes it is better suited for the Supplementary Information, we can move the table.

Responses to Reviewer #2:

1. Automation is important for making plasmid library. Please list pro and con of the traditional and Plasmidmaker in the discussion

Response:

We have revised the Discussion section by adding the following paragraph:

“Traditionally, multiple cloning procedures are performed manually, which is labor-intensive and error-prone. Specifically, the possibility of human induced errors becomes very high when one trying to clone a large batch of plasmids, where numerous pipetting steps with tiny volumes of materials are required. With automation of the cloning process, the assembly time is decreased tremendously as well as errors introduced by researchers can be avoided. In this work, $\sim 2 \times 10^4$ pipetting steps and transfer volume down to 2.5 nL are included to complete the automated plasmid construction. It will be very challenging to perform all these repetitive procedures manually with both precision and accuracy.”

2. Can the plasmidMaker assemble repetitive DNA sequences? This is generally very difficult for conventional plasmid library construction?

Response:

Yes, as long as the repetitive DNA sequence does not fall in the 24 bp *PfAgo*/AREs recognition sequence, plasmids with repetitive DNA sequences can be easily assembled using our method. In our guide design program, we also make sure that the recognition sequences for *PfAgo*/AREs are unique so that plasmids with repetitive DNA sequences can be assembled with high success rates. In the method development section, we show assembly of a 26.8 kb plasmid using seven DNA fragments with $\sim 63\%$ fidelity. This plasmid had four 500 bp repeated *TEF1* promoter sequences. All *I. orientalis* plasmids also contained multiple repeated sequences. For these plasmids, several promoters such as *TEF1*, *GPM1*, *g1414* are present in more than one copy in the assembled plasmids. All these plasmids also contained 3 copies of the same terminator *ENO2t*. Similarly, several plant plasmids contained either 2 or 3 copies of the *tNOS* terminator.

3. What are the largest and smallest sizes of plasmids that can be assembled by the PlasmidMaker?

Response:

The largest size we tried using our method is 27 kb with ~ 10 -fragments. Using our automated workflow, however, the largest size tried was 18 kb with 11 DNA fragments which we were able to successfully construct. For the smallest size, a minimal plasmid (~ 1.5 kb) containing an origin of replication and a resistance market can be assembled with our method. Smaller plasmid sizes have not been tested.

4. Since soft- and hardwares are needed for the PlasmidMaker, I assume ordinary labs cannot use the plasmidmaker now, am I right?

Response:

This software is a tool for users to easily construct and manipulate plasmids with their design goals. The software will be publicly available in the form of a user-friendly website, and users can design and upload their sequences on the website, just like using SnapGene or IDT to synthesize sequences. Users can also run the code on local machines by installing modules specified in the README file at https://github.com/ibiofab/PlasmidMaker_GuideDNA. The website will run the assembly pipeline in background and output all primers/guides required for this assembly. The hardware part is not necessary for doing the *PfAgo* based assembly in-house by ordinary labs. The ordinary labs can use the PlasmidMaker website to design the primers/guides for plasmid assembly. The website allows for download of primer/guide sequences. Then, those labs can order these oligos and perform the *PfAgo* based assembly in-house. Alternatively, the researchers can order the plasmids to be assembled from our automation facility iBioFAB. Currently, the ordering feature is under development for open access to all labs. In our initial phase, we have started offering the plasmid assembly service to researchers from the DOE Center for Advanced Bioenergy and Bioproducts Innovation (CABBI) starting February 2022. In our next phase, the iBioFAB facility will help build plasmids for other DOE bioenergy research centers, followed by opening it for ordering from all labs in the broad research community. The users can design the plasmids following guidelines available on our website. We will provide the users with a “status check” option to convey estimated assembly time and what stage of synthesis their plasmids are. Our technicians will use iBioFAB and fully utilize the complete PlasmidMaker software to assemble the ordered plasmids. Therefore, the users and other research labs do not need to possess all the hardware for plasmid construction.

Reviewers' Comments:

Reviewer #1:

Remarks to the Author:

The authors have resubmitted their paper "PlasmidMaker: a Versatile, Automated, and High Throughput End-to-End Platform for Plasmid Construction" with several changes based on my first review. These changes include (along with my comment on the change):

[Note the authors say "legend" throughout their response when they mean "caption". Adding an actual legend to the figures might actually have been valuable in some cases]

Added list of software environments used to make the tool

- This is good but still does not discuss the data structures and algorithms behind the tool. The good news is the code is open-source but I am trying to quickly understand where the innovation is. Why these languages were chosen is not motivated or how they fit into their larger vision of the open-source efforts they want to see.

"We will make the primer/guide design tool for PfAgo based assembly as a public website and a free online tool."

- This is key to the long-term impact of the tool. This should happen for the work to be published.

Modified Figure 1's caption to help with the input and output flow

- I don't find this particularly effective and I still believe the figure could have been re-organized. They make other claims about their desire to have the figure organized in its current form that I disagree with. I think it is easier to make a figure like the current one. I don't think it is better. Part "A" for example could have been made by any effort in this space with the only difference being a couple of text snippets. I think the paper's contribution to workflows in this space is diminished and less memorable due to this first initial figure.

Added Figure 2 text for clarity on the figure itself.

- Thank you for the changes. However, I still think this figure is a candidate for supplemental material. It would make more sense in the main manuscript if it were put more in the context of what the computational workflow does, where it falls in the workflow, and what challenges it solves. If figure #1 had been modified more, it might have allowed visual cues to be reused in this figure which placed this figure in the larger context of the whole paper.

Figure 3, "more clarifications, we added new text to both the main text and the Figure 3 legend to specify the 0-75% GC-content as well as referring the readers to the Supplementary Information on how this data was used for guide design."

- Again, this is a minimal edit that leaves the figure still being a bunch of graphs. I like the addition to the caption but now I have to find this point/area on all the graphs. I think all the raw data could have been put in the supplemental and instead have stylized data here that shows which portions of the graphs allow the authors to make their claims. Right now this figure just says to me "we did a lot of stuff".

Figure 4 also has material added to the caption.

- Currently, this figure suffers still from the same problems as Figure 3 (data with little indication of why it is included). Sentences however like "Based on these results, plasmid molecules with sizes up to 27 kb including the ones with 547 sequence repeats can be efficiently assembled by PfAgo/AREs generating 12 nt sticky ends from up to 10 548 DNA fragments" are REALLY KEY. Again, however, I have to read the entire caption just to get to this which could have been shown in the figure (why you are providing a visual presumably in the first place).

Figure 5 added loops and annotations.

- This looks good now and addressed my concerns.

Figure 6 added information about flowchart shapes to the caption

- Again more info is just put into the caption. Better than nothing but again, the impact could have been greater with easy additions to the visual.

Table 1

- Just responded to my concern in the response but did not make changes to the actual manuscript or table.

In general, I was disappointed that the text was largely just added into the captions for most of my feedback. Do not think the figures stand by themselves without repeated reading of both the caption and manuscript text. A paper published at this level should be better in this regard, in particular one about process and workflow.

Reviewer #2:

Remarks to the Author:

Authors have addressed my comments satisfactorily. I recommend to accept the paper.

We would like to thank the reviewers for their time and constructive comments. Below are our point-by-point responses to the reviewer #1's comments.

Reviewer #1 (Remarks to the Author):

The authors have resubmitted their paper "PlasmidMaker: a Versatile, Automated, and High Throughput End-to-End Platform for Plasmid Construction" with several changes based on my first review. These changes include (along with my comment on the change):

[Note the authors say "legend" throughout their response when they mean "caption". Adding an actual legend to the figures might actually have been valuable in some cases]

Added list of software environments used to make the tool

- This is good but still does not discuss the data structures and algorithms behind the tool. The good news is the code is open-source but I am trying to quickly understand where the innovation is. Why these languages were chosen is not motivated or how they fit into their larger vision of the open-source efforts they want to see.

Response:

Explanation for use of ReactJS and Django frameworks were added to the frontend software part in the main text. Here is the added section:

"We used industry-standard tools and practices for all phases of the software lifecycle. To facilitate rapid development, we selected the ReactJS framework for its widespread support for common open-source libraries and existing design templates. We selected the Django framework for backend processing for interoperability with existing laboratory software written in Python."

"We will make the primer/guide design tool for PfAgo based assembly as a public website and a free online tool."

- This is key to the long-term impact of the tool. This should happen for the work to be published.

Response:

The tool is now publicly available on our biofoundry website at <https://biofoundry.web.illinois.edu/>.

Modified Figure 1's caption to help with the input and output flow

- I don't find this particularly effective and I still believe the figure could have been re-organized. They make other claims about their desire to have the figure organized in its current form that I disagree with. I think it is easier to make a figure like the current one. I don't think it is better. Part "A" for example could have been made by any effort in this space with the only difference being a couple of text snippets. I think the paper's contribution to workflows in this space is diminished and less memorable due to this first initial figure.

Response:

To effectively highlight the innovation in this work, we integrated Figure 1 part b in the main workflow's build section to make it an all-in-one DBT diagram. Also, to make part a less generic as the reviewer pointed out, we highlighted the most important parts of our workflow with clear connection of outputs and inputs from each phase.

Added Figure 2 text for clarity on the figure itself.

- Thank you for the changes. However, I still think this figure is a candidate for supplemental material. It would make more sense in the main manuscript if it were put more in the context of what the computational workflow does, where it falls in the workflow, and what challenges it solves. If figure #1 had been modified more, it might have allowed visual cues to be reused in this figure which placed this figure in the larger context of the whole paper.

Response:

Although in previous review round, it was not mentioned that this figure is best suited for supplemental material, upon reviewer's request, we have moved this figure to the Supplementary Information document.

Figure 3, "more clarifications, we added new text to both the main text and the Figure 3 legend to specify the 0-75% GC-content as well as referring the readers to the Supplementary Information on how this data was used for guide design."

- Again, this is a minimal edit that leaves the figure still being a bunch of graphs. I like the addition to the caption but now I have to find this point/area on all the graphs. I think all the raw data could have been put in the supplemental and instead have stylized data here that shows which portions of the graphs allow the authors to make their claims. Right now this figure just says to me "we did a lot of stuff".

Response:

To address the clarification issues, we have moved Figure 3 parts b, c and d (raw data) to the Supplementary Information document and only included the average of data for both 9 and 12 nt sticky ends in the main figure. The conclusions were mainly based on these two graphs. In addition, for more clarity, we have added graph titles and more text descriptions in the figure itself.

Figure 4 also has material added to the caption.

- Currently, this figure suffers still from the same problems as Figure 3 (data with little indication of why it is included). Sentences however like "Based on these results, plasmid molecules with sizes up to 27 kb including the ones with 547 sequence repeats can be efficiently assembled by PfAgo/AREs generating 12 nt sticky ends from up to 10 548 DNA fragments" are REALLY KEY. Again, however, I have to read the entire caption just to get to this which could have been shown in the figure (why you are providing a visual presumably in the first place).

Response:

To make this figure clearer, we have moved the plasmid maps to the Supplementary Information document and only placed the graphs which were used for our conclusions in the main figure. For each graph, a title describing the experiment as well as additional text and legend were added to increase the clarity and make the figure stand by itself.

Figure 6 added information about flowchart shapes to the caption

- Again more info is just put into the caption. Better than nothing but again, the impact could have been greater with easy additions to the visual.

Response:

To address the reviewer's comment, we have simplified the figure to make the visual clearer. We have also enlarged the font and arrow sizes inside the figure to help readers understand our figure more easily.

Table 1

- Just responded to my concern in the response but did not make changes to the actual manuscript or table.

Response:

To address the reviewer's comment, besides the response we provided in the previous review round, we have made changes in the main text where Table 1 was mentioned.